# Spatial Structure Dynamics and Maintenance of a Natural Mixed Forest

**Chaofan Zhou** [1,2], **Di Liu** [1], **Keyi Chen** [3], **Xuefan Hu** [4], **Xiangdong Lei** [1], **Linyan Feng** [1], **Yuchao Zhang** [5] **and Huiru Zhang** [1,6,*]

1 Institute of Forest Resource Information Techniques, Chinese Academy of Forestry, Beijing 100091, China; cfzhou2021@163.com (C.Z.); liudi19920109@163.com (D.L.); xdlei@ifrit.ac.cn (X.L.); linyan_feng@caf.ac.cn (L.F.)
2 Ecology and Nature Conservation Institute, Chinese Academy of Forestry, Beijing 100091, China
3 Research Institute of Forestry Policy and Information, Chinese Academy of Forestry, Beijing 100091, China; lowrychen@sina.com
4 Beijing Institute of Landscape Architecture, Beijing 100102, China; hufanzi@163.com
5 Academy of Inventory and Planning, National Forestry and Grassland Administration, Beijing 100714, China; georgemacfee@163.com
6 Experimental Center of Forestry in North China, Chinese Academy of Forestry, Beijing 102300, China
* Correspondence: huiru@caf.ac.cn; Tel.: +86-10-69836348

**Abstract:** Spatial structure dynamics play a major role in understanding the mechanisms of forest structure and biodiversity formation. Recently, researches on the spatial structure dynamics utilizing multi-period data have been published. However, these studies only focused on comparative analyses of the spatial structure of multi-period living trees, without an in-depth analysis of the change processes. In this study, we propose a new comprehensive analysis method for dynamic change of the spatial structure at the individual level, which includes three processes (living trees' flow, mortality process and recruitment process) that have not been considered in previous researches. Four spatial structural parameters (SSSPs, Uniform angle index, Mingling, Dominance and Crowding) and a natural spruce-fir-broadleaf mixed forest with two-phase data were taken as an example to find out the laws of the spatial structure dynamics. All types of dynamic change were named and their proportions were analyzed. The proportion of changes in the SSSPs of individuals was relatively high, even though the mean values of the stand did not change considerably. The five values (0, 0.25, 0.5, 0.75, 1) of the SSSPs are in mutual flow, and the flows are typically one-step, with three-steps and four-steps changes being uncommon. The processes of mortality and recruitment have a higher influence on the spatial structure than the flow of living trees. The dynamic change of spatial structure analysis method created in this study can capture more features not discovered in earlier approaches, as well as guiding forest management in some ways. Understanding the nuances of these changes is a critical part of reasonable spatial structure and biodiversity maintenance, and should be the focus of future research efforts.

**Keywords:** spatial structure parameters; dynamic changes; the Sankey diagram; mortality process; recruitment process; natural mixed forests

## 1. Introduction

On long-term temporal and geographical dimensions, forest structure is the outcome of the combined activity of a range of ecological processes [1]. Complex forest structures form by diverse species, which in turn promotes the coexistence of multiple species [2]. Affected by the systems theory that "structure determines function" and the goal of diversity conservation, an increasing number of studies begin to focus on forest structure [3]. Spatial and non-spatial characteristics can be used to define the forest structure [3]. The indexes of non-spatial structure such as tree species composition [4], basal area [5], tree DBH

diversity [6,7] and stand density [8] represent the mean stand characteristics by neglecting the spatial information, while the spatial structure approaches take the distribution pattern of individuals and the spatial arrangement of their features into consideration; this includes stand crowding degree [9], species segregation index [10], DBH differentiation degree [11] and aggregation index [12]. It has been proved that spatial structure indicators play a greater role than non-spatial structure in stand reconstruction [13], describing the interaction between trees and thinning decision-making [9,14]. Especially the stand spatial structure parameters (SSSPs) [15], based on a structural unit constituted by a reference tree and its four nearest neighbors [13], have already been applied to the rationality evaluation of structural state [16,17], selective thinning [18,19] and close-to-nature planting for plantation [20].

The SSSPs, including Uniform angle index (*W*), Mingling (*M*), Dominance (*U*) and Crowding (*C*), can comprehensively analyze the micro spatial condition of individuals and the overall spatial structure of the stand [3,21]. These four parameters may be combined to generate a zero-variate (mean value), univariate, bivariate, trivariate and quadrivariate distribution, which can be used to evaluate forest structural properties at various resolutions [22]. Since it is simpler to understand and evaluate, zero-variate distribution, univariate distribution and bivariate distribution of parameters have been employed more commonly among them [18,23–29]. However, most of this research is concentrated on the spatial structure of forests in a specific time point. Response changes in spatial structure over time under anthropogenic and natural disturbances did not get the attention it deserves [30].

Fortunately, some scholars have noticed this, and several studies have been carried out. Deng and Katoh [31], Zhao et al. [32] and Xue et al. [33], and Zhang et al. [34] have analyzed the change of spatial structure characteristics on various natural forests and plantation in a short period (5 to 10 years) by the zero-variate, univariate and bivariate distribution methods, respectively. Wan et al. [35] used the zero-variate distribution and bivariate distribution of SSSPs to investigate the impacts of four treatments on the spatial structure dynamic of *Quercus aliena* var. *acuteserrata* forest in the Xiaolongshan mountain. These researchers found that the variations in the spatial structure are more affected by anthropogenic disturbances, especially structure-based forest management [35], than natural disturbances [36]. Unfortunately, the distribution dimensions (zero-variate to bivariate) of the spatial structure only had a slight effect on increasing the explanation of its change. All these approaches can examine changes in the spatial structure of living trees from early-stage to later-stage, though, it is unable to explain the origins of such changes adequately. As there are not only residual trees, but also felled trees, dead trees and recruitment trees during forest management or forest development [30,36], a simple examination of the difference in the spatial structure of live trees between the two phases cannot identify the causes of this change. This is not conducive to maintaining reasonable spatial structure through artificial measures. Therefore, a new method is needed to break the bottleneck.

Sankey diagrams represent the movement of information to and from various nodes in a network and are most commonly used to analyze energy and material flows [37]. These fluxes are represented by arrows or directed lines whose thickness corresponds to the size of the flow [38]. These diagrams are frequently used in industrial ecology to represent product lifetime evaluations, as well as in engineering to quickly visualize energy efficiency [37]. Sankey diagrams emphasize the quantity and direction of flows within a system, and they have been used in various geographic and human-environment study contexts due to their versatility [38]. The ability of Sankey diagrams to trace material movements has made them useful tools for estimating major greenhouse gas emissions [39,40], partitioning the global terrestrial water fluxes [41] and vegetation cover type conversion [42]. Although the use of the Sankey diagram in the study of spatial structure dynamics is innovative, it is ideally suited to tackling issues that cannot be answered using multidimensional distributions because it can accurately depict the changes between two periods. This study took the spruce-fir-broadleaf mixed forest as an example, disentangling its spatial structure

changes into "living trees' flow", "mortality process" and "recruitment process", aimed to: (1) develop a comprehensive methodology for spatial structure dynamics analysis based on SSSPs; (2) understand the laws in dynamic changes of spatial structure for spruce-fir-broadleaf mixed forest; (3) emphasize the important role of mortality process and recruitment process in the dynamic changes of spatial structure.

## 2. Materials and Methods

### 2.1. Study Area and Experimental Design

The research area is located in the Jingouling Forest Farm (130°5′–130°20′ E, 43°17′–43°25′ N), Wangqing Forestry Bureau, Jilin Province, China (Figure 1). It belongs to the middle and low mountains of Changbai Mountain, with an altitude between 300 and 1200 m and a slope between 5 and 25 degrees. The region has a temperate continental monsoon climate, with four distinct seasons of long winter and short summer, and the coexistence of rain and heat. The average annual temperature is 4 °C, and the annual average rainfall ranges from 600 to 700 mm. The area is dominated by grayish-brown soil with a moist and loose granular structure, which has an acidic pH and high fertility. Existing forest types are coniferous forest, broad-leaved forest and mixed forest.

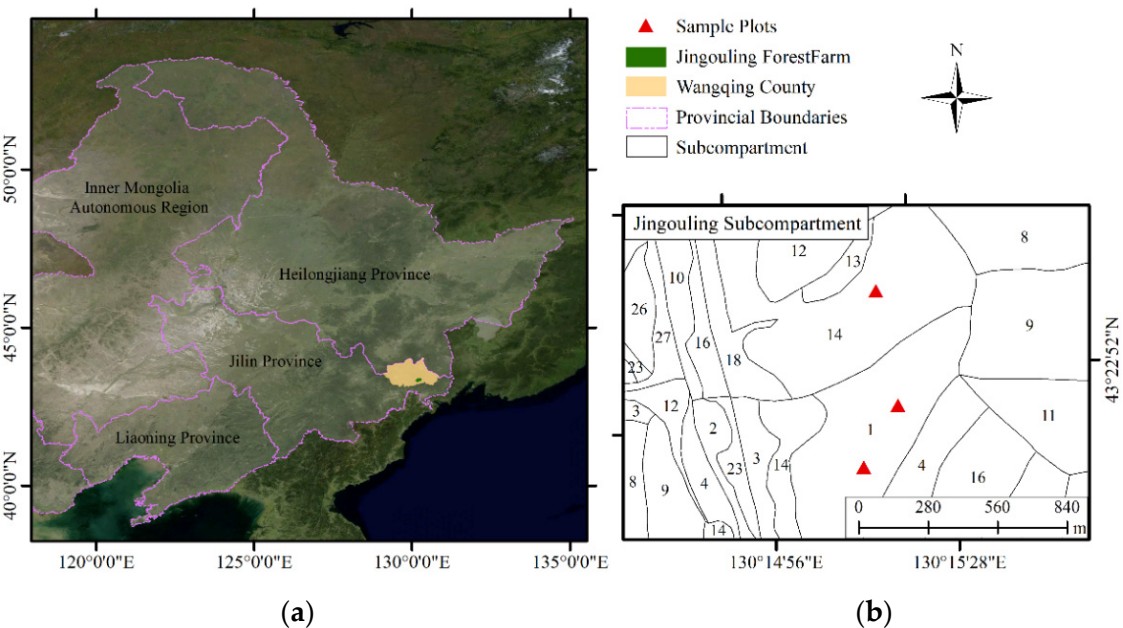

**Figure 1.** Location of the study area: Wangqing Forest Bureau (**a**) in northeast China and spatial distribution of 3 sample plots (**b**).

Three one-hectare sampling plots (100 m × 100 m each) were established in the natural spruce-fir-broadleaf mixed forest of the Jingouling Forest Farm in July 2013 (Figure 1). The main tree species in spruce-fir-broadleaf mixed forest include spruce (*Picea jezoensis* var. *microsperma* (Lindl.) Cheng et L.K. Fu), fir (*Abies nephrolepis* (Trautv.) Maxim.), larch (*Larix olgensis* Henry), Korean pine (*Pinus koraiensis* Siebold et Zucc.), white birch (*Betula platyphylla* Suk.), poplar (*Populus ussuriensis* Kom.), ribbed birch (*Betula costata* Trautv.), linden (*Tilia amurensis* Rupr.), elm (*Ulmus laciniata* (Trautv.) Mayr), maple (*Acer pictum* subsp. *mono* (Maxim.) H. Ohashi) and ash (*Fraxinus mandshurica* Rupr.). Trees with diameters at breast height (DBH) more than 5 cm were recorded, and species, DBH, tree height (H), crown diameter, relative coordinates and living state were measured (Table 1). In August 2018, these indicators were remeasured and recruitment trees (DBH of regeneration reach or exceed the threshold of DBH (5 cm) during the period of 2013–2018) were considered.

**Table 1.** Stand characteristics of 3 plots in 2013.

| Plot Code | Trees/ ha | Average DBH/cm | Basal Area/ (m²/ha) | Stock Volume/ (m³/ha) | Canopy Density | Composition of Tree Species |
|---|---|---|---|---|---|---|
| YLK1 | 996 | 17.63 | 24.30 | 199.97 | 0.85 | 2Bc2Ta1An1Pj1Pk1Lo1Pu1Am |
| YLK2 | 1024 | 18.23 | 26.72 | 216.00 | 0.86 | 2Bc2Ta1Lo1An1Pk1Am1Pj1Os |
| YLK3 | 1018 | 17.38 | 24.15 | 182.75 | 0.63 | 3Lo1Bc1Fm1Pk1An1Bp1Pj1Ta |

Note: Bp stands for *B. platyphylla*; Ta stands for *T. amurensis*; Bc stands for *B. costata*; Pk stands for *P. koraiensis*; An stands for *A. nephrolepis*; Lo stands for *L. olgensis*; Am stands for *A. pictum* subsp. *mono*; Fm stands for *F. mandschurica*; Pu stands for *P. ussuriensis*; Pj stands for *P. jezoensis*; Os stands for Other species.

*2.2. Stand Spatial Structure Parameters*

Four Stand Spatial Structure Parameters (SSSPs) including Uniform angle index, Mingling, Dominance and Crowding [3] were used to analyze the dynamics of the spatial structure of the spruce-fir-broadleaf mixed forest. Four parameters are calculated as follows:

1. Uniform angle index ($W$);

$$W_i = \frac{1}{4}\sum\nolimits_{j=1}^{4} z_{ij} \quad z_{ij} = \begin{cases} 1, & if\,\alpha_j < \alpha_0 \\ 0, & otherwise \end{cases} \tag{1}$$

In the Equation (1), $\alpha_j$ stands for the angle shown in and $\alpha_0$ stands for the standard angle 72°.

2. Mingling ($M$);

$$M_i = \frac{1}{4}\sum\nolimits_{i=1}^{4} v_{ij} \quad v_{ij} = \begin{cases} 1, & if\,species_j \neq species_i \\ 0, & otherwise \end{cases} \tag{2}$$

In the Equation (2), $species_j$ and $species_i$ denote the species of $jth$ neighboring tree and reference tree $i$, respectively.

3. Dominance ($U$);

$$U_i = \frac{1}{4}\sum\nolimits_{i=1}^{4} k_{ij} \quad k_{ij} = \begin{cases} 1, & if\,DBH_j \geq DBH_i \\ 0, & otherwise \end{cases} \tag{3}$$

In the Equation (3), $DBH_j$ and $DBH_i$ denote the diameter at breast height of $jth$ neighboring tree and reference tree $i$, respectively.

4. Crowding ($C$).

$$C_i = \frac{1}{4}\sum\nolimits_{j=1}^{4} y_{ij} \quad y_{ij} = \begin{cases} 1, & if\,c_j + c_i > dist_{ij} \\ 0, & otherwise \end{cases} \tag{4}$$

In the Equation (4), $c_j$ and $c_i$ stand for the crown radius of the $jth$ neighboring tree and reference tree $i$, respectively. The $dist_{ij}$ denotes the distance between the $jth$ neighboring tree and reference tree $i$. In a structural unit, these four SSSPs looked at the spatial and attribute correlations between the reference tree $i$ and its nearest neighbor $j$. Even though the four SSSPs focus on distinct parts of the spatial structure, they all have five possible values: 0, 0.25, 0.5, 0.75 and 1, which indicate the various spatial structure statuses (Figure 2). The quantization and analyses of SSSPs in this study were conducted using R version 4.1.1 [43].

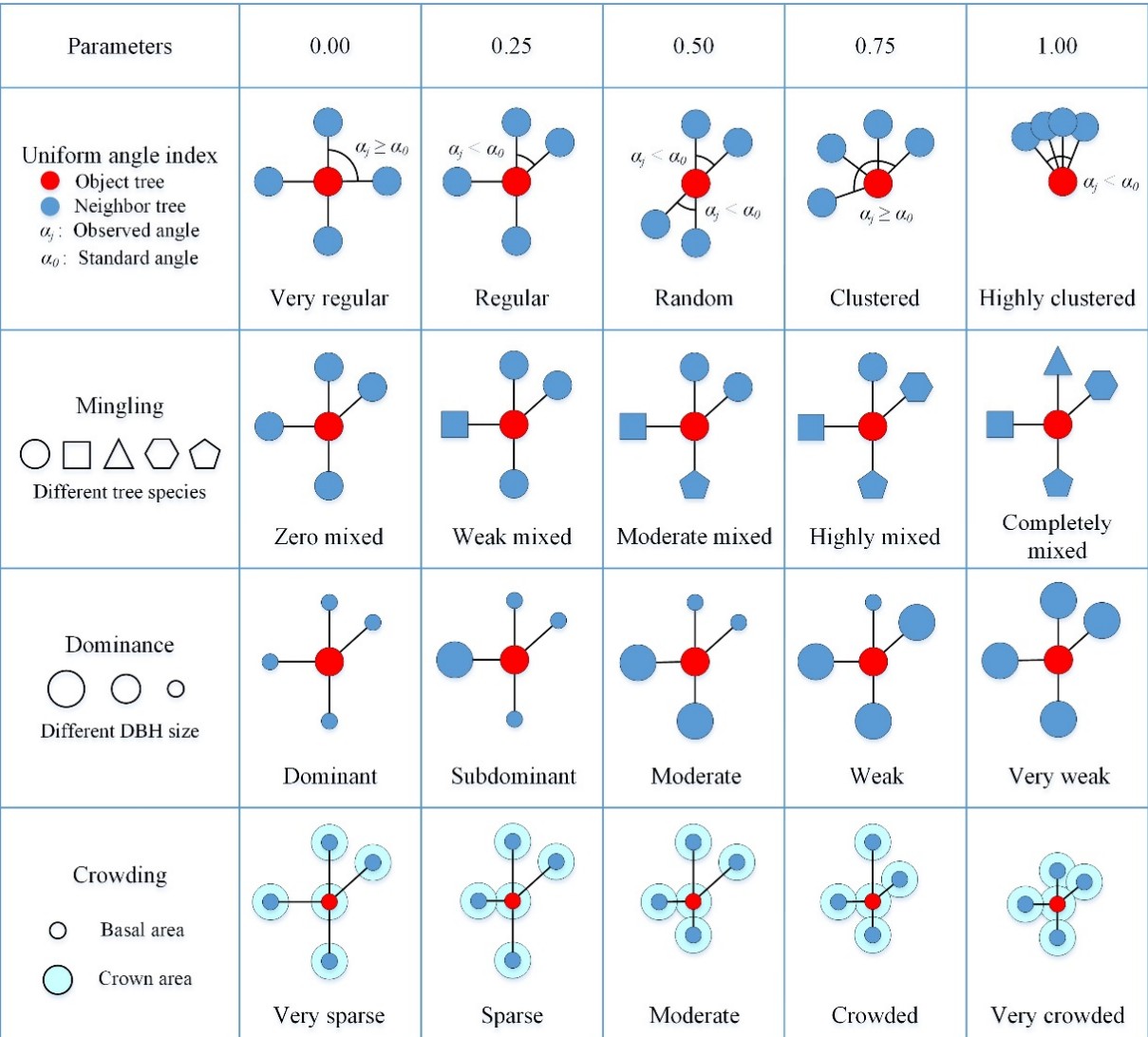

**Figure 2.** Schematic diagram of four stand spatial structural parameters. The Uniform angle index compared the size of observed angle $\alpha_j$ formed by two nearest neighbors and the reference tree to the standard angle $\alpha_0$ (72°); the Mingling, Dominance and Crowding compared whether the tree species are the same, the size of DBH and whether the crowns overlap between the reference tree and its four neighboring trees, respectively.

*2.3. Disentangle the Dynamic Changes of Spatial Structure*

In natural forests, the processes of mortality and recruitment are the primary causes of dynamic changes [30,36]. By incorporating these two processes into the dynamic change analysis of SSSPs, the source of changes can be better revealed. We used the ggalluvial [44] and ggplot2 [45] package in the R program [43] to create a Sankey diagram to show the flow change of SSSPs from early-stage to later-stage (Figure 3a), and all "change types" have been shown and named (Figure 3b), followed by statistics to further analyze the change law of SSSPs. The change law is evaluated primarily from three perspectives: the flow from living trees to living trees (*Flow*), the change from living trees to deadwood (*Dead*) and the conversion from regeneration to living trees (*Reg*), which respectively abbreviated as "living trees' flow", "mortality process" and "recruitment process".

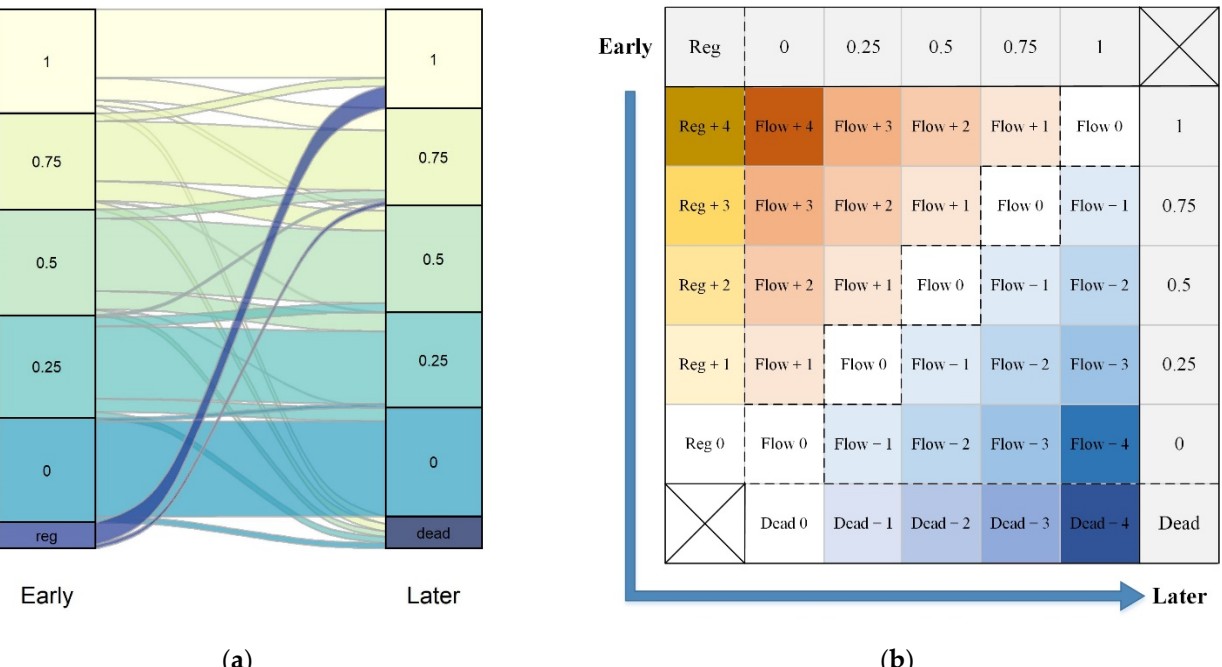

(**a**)                                                                                    (**b**)

**Figure 3.** The Sankey diagram (**a**) and Change types (**b**) of stand spatial structural parameters between two periods. The "reg" means the regeneration in early stage and growing into a living tree (DBH $\geq$ 5 cm) in later stage. The "dead" means the living trees in early stage, that have died in later stage. The flows in (**a**) connect the structural attributes of trees between the early and later periods. Each flow represents a change type shown in (**b**), and the width of the flow represents the relative proportion of this change type. The dotted lines in (**b**) divide 35 change types into recruitment process, ascending flow, stable flow, descending flow and mortality process with the colors of gold, orange, white, light blue and dark blue, respectively. The darker the color, the greater the change range of structure attributes. These change types in the ascending flow, stable flow and descending flow change can be further combined into nine change types according to the change range (the depth of color, see Section 2.3.1 for more details).

2.3.1. The Flow from Living Tree to Living Tree

Each SSSPs have five possible values: 0, 0.25, 0.5, 0.75 and 1. There are three flow directions of values of living trees from the early-stage to the later-stage when the values are placed from top to bottom in descending order as in Figure 3a: "ascending", "stable" and "descending". A shift of one-step, such as from 0 to 0.25 or 1 to 0.75, is referred to as a "one-step change". When combining flow direction and step, the case like the value changes from 0 and 1 is called "ascending four-steps". On the contrary, when called "descending four-steps", while the value that does not change is called "stable". The naming method of other value changes are analogized.

Based on the naming method, there are nine types of living trees flow of SSSPs, namely "ascending four-steps (Flow + 4)", "ascending three-steps (Flow + 3)", "ascending two-steps (Flow + 2)", "ascending one-step (Flow + 1)", "stable (Flow 0)", "descending one-step (Flow − 1)", "descending two-steps (Flow − 2)", "descending three-steps (Flow − 3)" and "descending four-steps (Flow − 4)". The number of corresponding types increased from 1 to 5 and then decreased to 1 (Figure 3b). As shown by the dashed line in Figure 3b, the nine types of living trees' flow can be divided into three parts, the "ascending flow ($Flow_{Up}$)" in the upper left triangle (Flow + 4, Flow + 3, Flow + 2, Flow + 1), the "stable flow ($Flow_{Stable}$)" in the middle diagonal (Flow 0), and the "descending flow ($Flow_{Down}$)" in the lower right triangle. (Flow − 1, Flow − 2, Flow − 3, Flow − 4).

### 2.3.2. The Change from Living Tree to Deadwood

When trees die for a reason internal or external, then their SSSPs will change from five possible values to no value (just like the impact of value 0 when computing the mean value of SSSPs). As a result, the naming type for the mortality process is identical to that of 2.3.1, with names like "Dead 0", "Dead − 1", "Dead − 2", "Dead − 3" and "Dead − 4" (see Figure 3b).

### 2.3.3. The Conversion from Regeneration to Living Trees

The regeneration (seeding and sapling) first contributed to the computation of spatial structure when its DBH is more than 5 cm (becoming a living tree), then their SSSPs will change from no value to five possible values (opposite to the mortality process). Accordingly, the type naming for the recruitment process is analogical to that of 2.3.1, with names like "Reg 0", "Reg + 1", "Reg + 2", "Reg + 3", "Reg + 4" (see Figure 3b).

By calculating the proportion of each change type in these three processes, we can comprehend the impact of each process on the overall spatial structure and analyze the stability of SSSPs at the individual, species and stand level.

### 2.4. Equation of Changes in Spatial Structure

By disentangling the process of spatial structure change, we assume that the relationship between the mean values of SSSPs of early-stage and later-stage conforms to the following equation:

$$\omega_{Early} \times N_{Early} + Flow_{Up} + Flow_{Stable} + Flow_{Down} + Dead + Reg = \omega_{Later} \times N_{Later} \quad (5)$$

where, $\omega_{Early}$, $\omega_{Later}$, $N_{Early}$, $N_{Later}$ stand for the mean values of SSSPs and the number of trees in the early and late periods, respectively. $Flow_{Up}$, $Flow_{Stable}$, $Flow_{Down}$, $Dead$ and $Reg$ denote the changes of SSSPs caused by ascending flow, stable flow, descending flow, mortality process and recruitment process, respectively. The calculation formulas are as follows:

$$\begin{cases} Flow_{Up} = 1 \times N_{Flow+4} + 0.75 \times N_{Flow+3} + 0.5 \times N_{Flow+2} + 0.25 \times N_{Flow+1} \\ Flow_{Stable} = 0 \times N_{Flow0} \\ Flow_{Down} = -0.25 \times N_{Flow-1} - 0.5 \times N_{Flow-2} - 0.75 \times N_{Flow-3} - 1 \times N_{Flow-4} \\ Reg = 1 \times N_{Reg+4} + 0.75 \times N_{Reg+3} + 0.5 \times N_{Reg+2} + 0.25 \times N_{Reg+1} + 0 \times N_{Reg0} \\ Dead = -0 \times N_{Dead0} - 0.25 \times N_{Dead-1} - 0.5 \times N_{Dead-2} - 0.75 \times N_{Dead-3} - 1 \times N_{Dead-4} \end{cases} \quad (6)$$

$N_i$ stands for the number of trees of the *i*-th change type mentioned above, and the relationship between $N_{Early}$ and $N_{Later}$ is as follows:

$$\begin{cases} N_{Early} + N_{Reg} - N_{Dead} = N_{Later} \\ N_{Reg} = N_{Reg+4} + N_{Reg+3} + N_{Reg+2} + N_{Reg+1} + N_{Reg0} \\ N_{Dead} = N_{Dead0} + N_{Dead-1} + N_{Dead-2} + N_{Dead-3} + N_{Dead-4} \end{cases} \quad (7)$$

Both the ascending flow ($Flow_{Up}$) and the recruitment process ($Reg$) will increase the sum of the SSSPs' values ($\omega_{Later} \times N_{Later}$) in the later-stage, while the descending flow ($Flow_{Down}$) and the mortality process ($Dead$) are the opposite, and the stable flow ($Flow_{Stable}$) does not change it. The difference between $N_{Reg}$ and $N_{Dead}$ determines whether $N_{Later}$ increases or decreases compared with $N_{Early}$, which in turn affects the mean values of SSSPs of later-stage. To validate the feasibility of the correlation formula between the mean values of the SSSPs early and later, it must be extensively tested with actual samples (see below).

## 3. Results

### 3.1. The Changing Process of Stand Spatial Structure in Early- and Later-Period

The distribution proportions of the other three SSSPs rarely changed from 2013 to 2018, except for modest variations in the distribution proportions of Crowding (Figure 4).

The changing process of stand spatial structure will be analyzed in the following three (Sections 3.1.1–3.1.3), respectively.

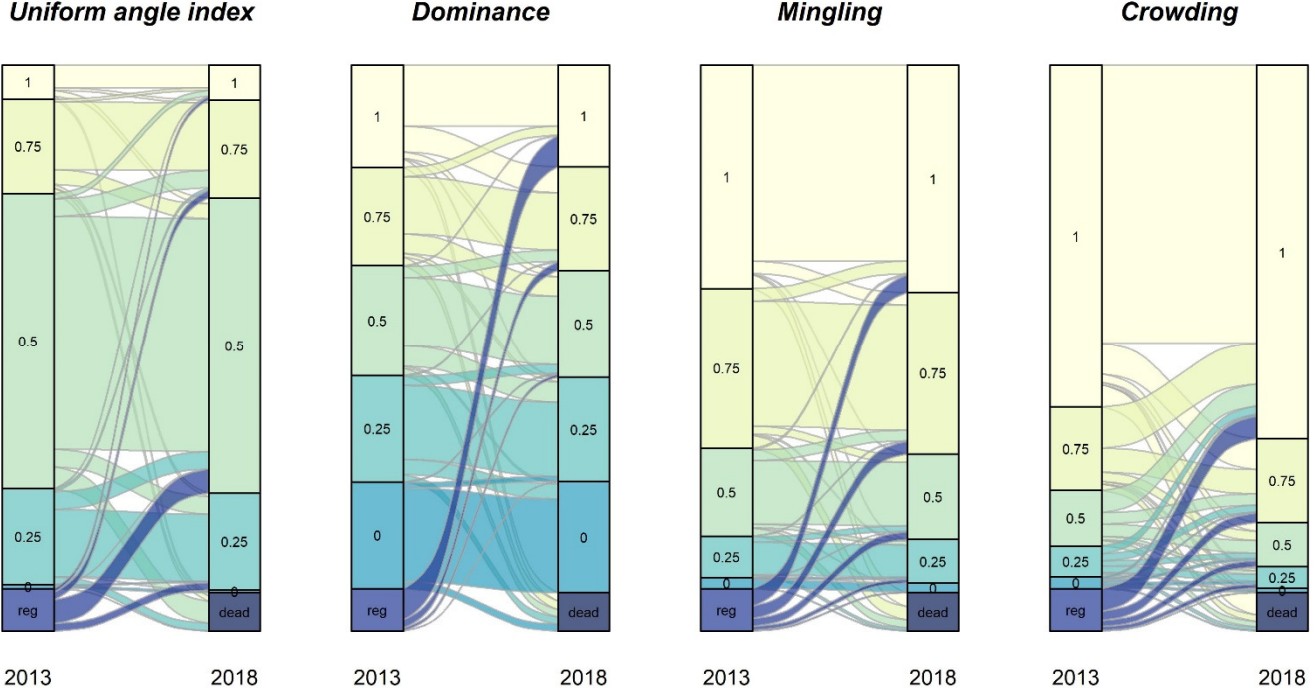

**Figure 4.** Changes in values distribution of different stand spatial structural parameters from 2013 to 2018 of all three plots. The "reg" means the regeneration in 2013 and growing into a living tree (DBH ≥ 5 cm) in 2018. The "dead" means the living trees in 2013, that have died in 2018. The flows in the figure connect the structural attributes of trees between 2013 and 2018. Each flow represents a change type of structure attributes, and the width of the flow represents the relative proportion of this change type.

### 3.1.1. The Flow of Living Trees

During the natural development of a spruce-fir-broadleaf mixed forest, there is almost no "four-steps change" (0–1.3%), and very little "three-steps change" (0–5%) and "two-steps change" (0–15.3%), while the "one-step change" are relatively large (3.7%–27%), second only to the case where the value does not change (46.7%–89.2%) (Figure 4 and Table 2). In most circumstances, changes between two values are not unidirectional, but rather flow in both directions (ascending and descending coexist), however, the quantity of mutual flow may be unequal.

**Table 2.** The proportion statistics of change types in living trees' flow (%).

| Types | Uniform Angle Index | Dominance | Mingling | Crowding |
|---|---|---|---|---|
| Flow + 4 | 0.0 ± 0.0 (0.0–0.0) | 0.0 ± 0.0 (0.0–0.0) | 0.0 ± 0.0 (0.0–0.0) | 0.4 ± 0.4 (0.0–1.3) |
| Flow + 3 | 0.1 ± 0.0 (0.0–0.1) | 0.0 ± 0.0 (0.0–0.0) | 0.0 ± 0.0 (0.0–0.0) | 1.7 ± 1.4 (0.0–5.0) |
| Flow + 2 | 2.0 ± 0.3 (1.5–2.9) | 0.6 ± 0.2 (0.3–0.9) | 0.6 ± 0.0 (0.5–0.7) | 6.2 ± 3.7 (1.4–15.3) |
| Flow + 1 | 8.3 ± 0.3 (7.8–8.9) | 6.7 ± 0.3 (6.1–7.3) | 6.6 ± 0.4 (5.7–7.2) | 12.5 ± 5.9 (3.7–27.0) |
| Flow 0 | 80.3 ± 1.2 (77.7–82.5) | 73.6 ± 2.3 (68.9–78.8) | 87.3 ± 0.9 (85.4–89.2) | 66.7 ± 8.2 (46.7–76.9) |
| Flow − 1 | 7.4 ± 0.7 (5.7–8.4) | 16.2 ± 1.9 (13.1–20.7) | 4.9 ± 0.6 (3.9–6.4) | 9.6 ± 2.2 (4.5–13.6) |
| Flow − 2 | 1.9 ± 0.4 (1.2–2.7) | 2.7 ± 0.5 (1.7–3.6) | 0.5 ± 0.3 (0.0–1.1) | 2.3 ± 1.0 (0.0–4.2) |
| Flow − 3 | 0.0 ± 0.0 (0.0–0.0) | 0.2 ± 0.1 (0.0–0.4) | 0.0 ± 0.0 (0.0–0.1) | 0.5 ± 0.2 (0.1–0.9) |
| Flow − 4 | 0.0 ± 0.0 (0.0–0.0) | 0.0 ± 0.0 (0.0–0.0) | 0.0 ± 0.0 (0.0–0.0) | 0.0 ± 0.0 (0.0–0.1) |

Note: Values are presented as the mean ± standard error (minimum–maximum).

Analysis of the proportion of Flow 0 (value does not change) of each SSSPs shows that Mingling, followed by Uniform angle index and Dominance, is the steadiest, while

Crowding is the least stable (Table 2). For Uniform angle index, Dominance, Mingling and Crowding, the most stable and the least stable tree species are larch and poplar, white birch and fir, poplar and fir and poplar and larch, respectively (Table A1 in Appendix A).

When the fraction of ascending and descending change types is compared, it is clear that the overall value of Uniform angle index, Mingling and Crowding grew while Dominance fell throughout the living trees' flow (Table 2). The shifting trends of parameters for most tree species are the same as those of stands, with the exception of the following exceptions. Species including white birch, linden, ribbed birch, and poplar declined in the Uniform angle index; birch increased in Dominance; fir, maple, ash, and spruce reduced in Mingling; and fir decreased in Crowding (Table A1).

### 3.1.2. The Process of Mortality

Compared with the corresponding values in 2013, the proportion of change types in Uniform angle index (Dead − 2 and Dead − 3), Dominance (Dead 0, Dead − 1 and Dead − 2), Mingling (Dead 0, Dead − 1 and Dead − 4), Crowding (Dead − 4) is lower (Table 3), indicating that the mortality process encourages an increase in the number of reference trees with the following situation: random and aggregated of Uniform angle index ($W$ = 0.5 and 0.75), dominant, subdominant and moderate of Dominance ($U$ = 0, 0.25 and 0.5), zero mixed, weak mixed and extremely highly mixed of Mingling ($M$ = 0, 0.25 and 1), very crowded of Crowding ($C$ = 1).

**Table 3.** The proportion statistics of change types in mortality process (%).

| Types | Uniform Angle Index | Dominance | Mingling | Crowding |
|---|---|---|---|---|
| Dead 0 | 1.0 ± 0.4 (0.0 − 1.7) | 17.6 ± 1.5 (15.0–21.1) ↓ | 2.3 ± 0.6 (1.4–3.7) ↓ | 2.3 ± 1.4 (0.0–5.6) |
| Dead − 1 | 22.6 ± 5.2 (9.9–29.6) | 19.9 ± 0.6 (18.5–21.1) ↓ | 7.5 ± 2.5 (1.4–11.1) ↓ | 8.4 ± 5.6 (1.4–22.2) |
| Dead − 2 | 56.4 ± 2.1 (51.9–60.6) ↓ | 20.0 ± 2.1 (16.7–25.0) ↓ | 22.5 ± 3.6 (14.8–30.0) | 14.6 ± 3.9 (8.3–24.1) |
| Dead − 3 | 13.2 ± 2.4 (8.3–18.3) ↓ | 21.2 ± 0.6 (19.7–22.2) | 34.2 ± 0.7 (32.4–35.2) | 17.9 ± 3.3 (12.7–25.9) |
| Dead − 4 | 6.8 ± 1.3 (5.0–9.9) | 21.3 ± 1.9 (18.3–25.9) | 33.6 ± 4.5 (23.3–42.3) ↓ | 56.8 ± 14.1 (22.2–75.0) ↓ |

Note: Values are presented as the mean ± standard error (minimum–maximum). "↓" indicated that the proportion of this change type is smaller than corresponding values in 2013.

Some tree species exhibit significant variances with the stand in change patterns of four parameters throughout the mortality process, as shown in Table A2: White birch, linden, ash, poplar, and elm have increased in Uniform angle index ($W$ = 1), whereas fir and spruce have increased in Dominance ($U$ = 0.75 and 1), respectively. The Mingling of linden, larch, poplar, and spruce, zero mixed and weak mixed ($M$ = 0 and 0.25) have not risen; very crowded ($C$ = 1) have not increased in Crowding of linden and elm.

### 3.1.3. The Process of Recruitment

Compared with the corresponding values in 2018, the proportion of change types in Uniform angle index (Reg + 1, Reg + 3 and Reg + 4), Dominance (Reg + 3 and Reg + 4), Mingling (Reg + 1, Reg + 2 and Reg + 3) and Crowding (Reg + 1, Reg + 2, Reg + 3 and Reg + 4) is larger (Table 4), showing that the recruitment process promotes a rise in the number of reference trees with the following situation: regular, clustered and very clustered of Uniform angle index ($W$ = 0.25, 0.75 and 1), weak and very weak of Dominance ($U$ = 0.75 and 1), weak mixed, moderate mixed and highly mixed of Mingling ($M$ = 0.25, 0.5 and 0.75), very sparse, sparse, moderate and crowded of Crowding ($C$ = 0, 0.25, 0.5 and 0.75).

The white birch, ribbed birch, poplar, Larch, and Ash have no or only one tree in the recruitment process, but the fir accounts for more than half of the population in the recruitment. Although the change patterns of four stand characteristics are nearly equivalent to fir, several tree species differ significantly in the following circumstances (Table A3): random ($W$ = 0.5) has grown in Uniform angle index of Korean pine, maple, elm, and spruce; highly mixed or completely mixed ($M$ = 0.75 and 1) has increased in Mingling of Korean pine,

maple, ash, elm, and spruce; and very crowded (*C* = 1) has increased in Crowding of elm and spruce.

**Table 4.** The proportion statistics of change types in recruitment process (%).

| Types | Uniform Angle Index | Dominance | Mingling | Crowding |
|-------|--------------------|-----------|----------|----------|
| Reg 0 | 0.0 ± 0.0 (0.0–0.0) | 0.0 ± 0.0 (0.0–0.0) | 2.2 ± 1.4 (0.0–5.6) | 1.2 ± 0.5 (0.0–2.2)↑ |
| Reg + 1 | 19.2 ± 3.4 (11.0–23.6)↑ | 0.5 ± 0.4 (0.0–1.4) | 6.8 ± 3.3 (0.0–13.9)↑ | 7.1 ± 0.8 (5.1–8.3)↑ |
| Reg + 2 | 55.2 ± 2.8 (48.6–60.4) | 6.0 ± 3.0 (0.0–12.5) | 16.3 ± 6.0 (2.6–27.8)↑ | 13.6 ± 3.0 (9.7–20.9)↑ |
| Reg + 3 | 19.2 ± 0.7 (17.9–20.8)↑ | 20.8 ± 0.6 (19.8–22.2)↑ | 32.2 ± 5.1 (22.2–43.6)↑ | 22.5 ± 1.1 (20.5–25.0)↑ |
| Reg + 4 | 6.5 ± 1.7 (2.6–9.9)↑ | 72.7 ± 3.8 (63.9–79.5)↑ | 42.4 ± 5.5 (30.6–53.8) | 55.6 ± 4.0 (47.3–64.1) |

Note: Values are presented as the mean ± standard error (minimum–maximum). "↑" indicated that the proportion of this change type is larger than corresponding values in 2018.

### 3.2. The Variation and Disentangle of Mean Values of Stand Spatial Structure Parameters

The shifting patterns of the four SSSPs of the three plots were not consistent from 2013 to 2018 (Figure 5). The mean values of Uniform angle index and Mingling in all three plots tended to rise, indicating that the degree of tree aggregation and intermixing of tree species in the plots has increased. The mean values of Crowding in YLK1 and YLK2 decreased with time, showing these plots became less crowded, whereas the converse was true for plot YLK3. The mean of Dominance changes is complicated and has no fixed trend (Figure 5). In the average variation range, Crowding (0.090) > Uniform angle index (0.006) > Mingling (0.006) > Dominance (0.004).

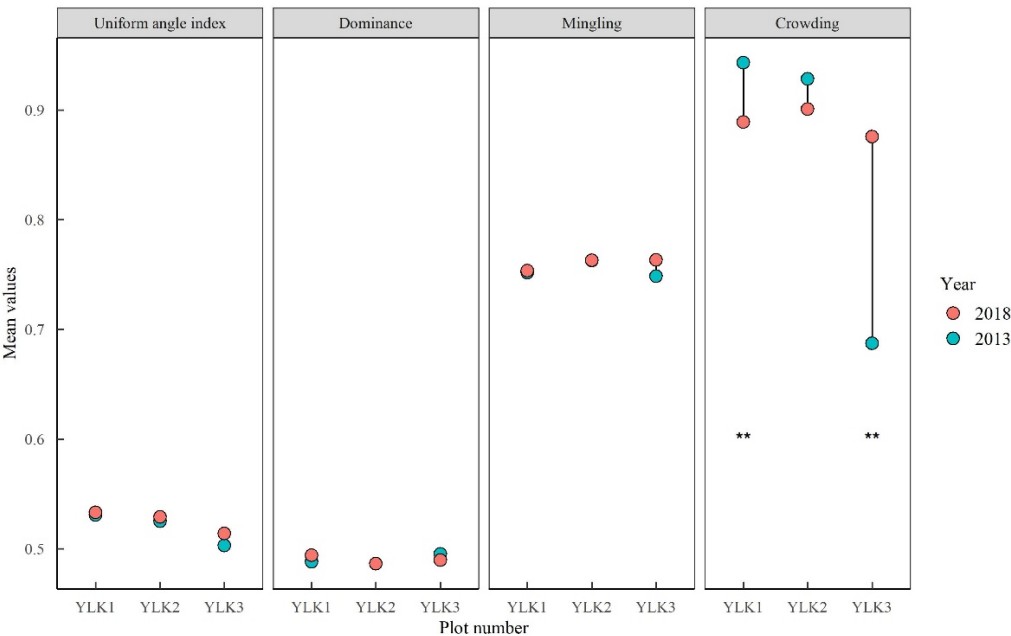

**Figure 5.** Changes in mean values of stand spatial structural parameters of plots from 2013 to 2018. The Chi-square test was used to check the significant difference between years for each of the four parameters of the five possible values (0, 0.25, 0.5, 0.75 and 1). Significant codes: $p < 0.01$ '**'.

The total value of the previous spatial structure (the product of the mean value of SSSPs and the number of trees in the early-stage ($\omega_{early} \times N_{early}$)), plus the sum of the five parts of SSSPs change processes at the individual level ($Flow_{up}$, $Flow_{stable}$, $Flow_{down}$, *Dead* and *Reg*), are exactly equal to the sum of the later spatial structure values (the product of the mean value of the SSSPs and the number of trees in later-stage($\omega_{later} \times N_{later}$)), irrespective of the category of SSSPs and sample plots (Table 5). The values of *Dead* or *Reg* are generally greater than that of $Flow_{up}$ or $Flow_{down}$ for the sum (absolute value) of the value changes of the SSSPs. In Dominance, the values of *Reg* are invariably greater than that of *Dead*, and the

values of $Flow_{down}$ are always bigger than that of $Flow_{up}$, whereas other SSSPs have little difference or are inconclusive.

**Table 5.** Disentangle the changes in spatial structure. The values in the table equal to the sum of the product the type value and quantity of SSSPs (see equations in Section 2.4). Positive and negative values in the change processes indicate an increase and decrease in the total value of SSSPs from early-stage to later-stage, respectively.

| Parameters | No. | Early-Stage | Change Processes | | | | | Later-Stage |
|---|---|---|---|---|---|---|---|---|
| | | $\omega_{early} \times N_{early}$ | $Flow_{up}$ | $Flow_{stable}$ | $Flow_{down}$ | $Dead$ | $Reg$ | $\omega_{later} \times N_{later}$ |
| Uniform angle index | YLK1 | 427.25 | 26.25 | 0.00 | −25.50 | −40.00 | 51.75 | 439.75 |
| | YLK2 | 420.00 | 20.75 | 0.00 | −21.25 | −28.00 | 38.00 | 429.50 |
| | YLK3 | 405.75 | 23.25 | 0.00 | −15.25 | −26.25 | 19.50 | 407.00 |
| Dominance | YLK1 | 393.00 | 14.50 | 0.00 | −49.25 | −34.75 | 84.00 | 407.50 |
| | YLK2 | 389.00 | 17.00 | 0.00 | −42.75 | −31.25 | 62.75 | 394.75 |
| | YLK3 | 399.50 | 12.50 | 0.00 | −31.25 | −30.00 | 37.00 | 387.75 |
| Mingling | YLK1 | 605.00 | 15.25 | 0.00 | −13.25 | −55.50 | 70.00 | 621.50 |
| | YLK2 | 610.00 | 15.25 | 0.00 | −12.00 | −40.25 | 46.50 | 619.50 |
| | YLK3 | 604.00 | 13.25 | 0.00 | −8.25 | −38.75 | 34.25 | 604.50 |
| Crowding | YLK1 | 759.25 | 11.75 | 0.00 | −43.75 | −63.00 | 69.25 | 733.50 |
| | YLK2 | 742.75 | 19.75 | 0.00 | −35.00 | −54.50 | 58.50 | 731.50 |
| | YLK3 | 554.50 | 146.75 | 0.00 | −9.25 | −32.00 | 33.50 | 693.50 |

## 4. Discussion

### 4.1. The Dynamics of Spatial Structure and Its Implications for Management

Continuous spatial structure observations for individual trees were not carried out in previous studies, and ongoing research mainly focused more on changes at the stand level, such as the mean value or distribution of SSSPs. However, the changes in the mean value are generally slight. By following the change of the spatial structure at the individual level, the changes in the stand spatial structure can be disentangled into three categories to better understand the change process: the living trees' flow, the mortality process and the recruitment process. The formation of the spatial structure variation equation was validated during the process by comparing the mean values of the SSSPs of the sample plots in early-stage and later-stage, and the accuracy of disentangling process was confirmed. We were able to achieve several outcomes that were not found using typical approaches thanks to the methods used in this research:

(a) The proportion of changes in the spatial structure of individuals was relatively high, even though the overall mean values of the stand did not change considerably. This indicates that the changes in spatial structure at the individual level have a large counteracting effect when summarized to stand level. In order to further study the changes in spatial structure, it is necessary to start from the dynamics of spatial structure at the individual level.

(b) The five values of the SSSPs are in mutual flow, although the flow may be asymmetric. Flow changes are typically one-step, with three-steps and four-steps changes being uncommon. It is suggested that the change of spatial structure is a gradual process during the development of the stand, which has some guidance for imitating the natural change of stand spatial structure in forest management.

(c) In general, the processes of mortality and recruitment have a higher influence on the spatial structure than the flow of living trees. This is due to the dead woods and recruitment trees not only changing the stand spatial structure through their values, but also changing the spatial structure of other reference trees which take them as adjacent trees. A further thorough analysis of the influence mechanism on stand spatial structure can be explored through the mortality process and recruitment process.

(d)   The impacts of the mortality and recruitment process on the direction of spatial structure variety are opposites, but their influences are equivalent in most cases. This suggests that the role of both cutting (mortality process) and replanting (recruitment process) measures should be emphasized in improving and maintaining the spatial structure [18,35].

We also found that the mean values of Uniform angle index and Mingling of the plots increased, whereas Crowding and Dominance both increased and decreased in five years. Our results share several similarities and differences with Deng and Katoh [31], Zhao et al. [32], Xue et al. [33] and Wan et al. [35], which may be due to the correlation between the change direction of the SSSPs in different forest communities and the initial spatial structure at the stand level.

According to Hui et al. [46], the mean value of Uniform angle index should remain at (0.475, 0.517), indicating an acceptable pattern of random distribution. From 2013 to 2018, the mean values of the Uniform angle index of three plots all rose, and YLK1 and YLK2 have already crossed the upper threshold of the tolerable interval, with YLK3 perhaps exceeding the upper threshold if the development trend continues ($W = 0.75$ and 1). Given this, in the reference unit with clustered and very clustered conditions, one or two clustered neighboring trees in the unit should be removed, and the opposite way should be checked for an adjacent tree to reintegrate the unit and decide the implementation of replanting [47].

In general, trees in a dominant condition ($U = 0$ and 0.25) have a better chance of surviving, and our results at the stand level back this up. However, other species, such as fir and spruce, are not affected in our study, dominant fir and spruce died more frequently than tiny ones (a weak state with $U = 0.75$ and 1). This might be explained by these species' shallow roots, which are readily blown down by the wind or broken down by the snow [48,49]. As a result, thinning should be done away from these species' huge trees to ensure that nearby trees provide adequate support.

The variety of tree species is frequently advocated, particularly spatial mixing, because pest and disease propagation can be reduced if the species are mixed [50]. In our study, the mean values of Mingling of spruce-fir-broadleaf mixed forest are steady at approximately 0.75 or slightly increased; most tree species show the same trend, although fir, maple, ash, and spruce have declined. One possible explanation is that the recruiting tree with the same species reintegrated the unit of the reference tree (such as the species often grow into a clump). Trees with Mingling descending flow and states of zero mixed, weak mixed, and moderate mixed ($M = 0$, 0.25, and 0.5) should be examined for thinning for enhancing Mingling, meanwhile the thinning should also guide by the health condition and dominance. [18].

Crowding has increased over time induced by tree canopy development in most species and plot YLK3, whereas reduced in fir and plots YLK1 and YLK2. This suggested that plots YLK1 and YLK2 had self-thinning due to the intense crown competition, and thinning procedures should be implemented as soon as feasible. It may also reveal that the nearby tree is more possible to die in the unit of reference tree of fir.

According to the aforementioned analysis, the fir should be improved first in the spruce-fir-broadleaf mixed forest of our study, followed by maple, ash and spruce; thinning should highlight Crowding, with the help of Mingling and Uniform angle index; and, if required, Mingling and Uniform angle index should also guide the species and the position of replanting.

*4.2. The Reasons for the Change of Stand Spatial Structure Parameters*

According to the construction method of the SSSPs, the reasons for the change of the spatial structure at the individual level can be attributed to:

(a)   The reference tree dies or recruitment, causing the spatial structure value change from existence to nothing or in the opposite;

(b)   The death or recruitment of the neighbors of the reference tree induces structural unit rearrangement [47], which alters the size, species, relative location, and canopy overlap relationships between the reference tree and its neighbors;

(c)   A qualitative change has occurred in the connection between the size and canopy overlap of neighboring trees and the reference tree.

All four SSSPs could be affected by (a) and (b), whereas Dominance and Crowding are also affected by (c). The main reason for the change of the spatial structure is (c) when there is no change in the composition of a structural unit formed by the reference tree and neighboring trees. Otherwise, the main reason is (b), and (c) plays a supporting role. In a developing forest with tree mortality and recruitment, it can be shown that the change of spatial structure is generally dominated by tree mortality and recruitment processes, whilst the tree growth process just fine-tunes the change of spatial structure. The research on natural spruce-fir-broadleaf mixed forest backs this up.

The change of mean values at the stand level can be explained by the change in the values of the SSSPs at the individual level, according to the above theory and our study data. The stability of SSSPs at the individual level may be determined using the ratio of Flow 0 in the flow of living trees of a natural spruce-fir-broadleaf mixed forest. The most stable parameter is Mingling, which is followed by Uniform angle index, Dominance, and lastly Crowding.

The Dominance and Crowding in living trees' flow are affected by (b) and (c), so the stabilities are not as high as Mingling and Uniform angle index stability, which are solely affected by (b). The Dominance is more stable than Crowding, which could be explained as the replacement of adjacent trees of the reference tree caused by (b) has an equivalent effect on both the Dominance and Crowding. However, the change in canopy overlap relationship caused by (c) is greater than the change in tree size connection between reference tree and its neighbors, so the effect on Crowding is greater. The Mingling is more stable than the Uniform angle index, which might be owing to the fact that reference trees account for up to 71.4 percent of highly mixed and completely mixed ($M = 0.75$ and 1), and the replacement of neighboring tree species induced by (b) did not modify the state of mixing intensity of the reference tree, whereas the replacement of adjacent trees could cause a significant change in the positional relation between reference tree and its neighbors, resulting in a change in the Uniform angle index.

The mean values of Crowding fluctuate a lot, the Mingling and Uniform angle index are moderate, and the Dominance is most stable. According to Deng and Katoh [31], Zhao et al. [32], Xue et al. [33] and Wan et al. [35], the variation of the Mingling was larger than that of the Uniform angle index, while Deng and Katoh [31] and Xue et al. [33] calculated the mean of the Dominance and found that the variation of the Mingling was also larger than Dominance and Uniform angle index is equivalent with Dominance at most of the time.

The Dominance is determined by comparing the size relationships of the reference tree and its neighbors [3]. Since each tree in the core area is not only a reference tree, but also an adjacent tree of other trees, the Dominance will be around 0.5 when aggregated to the mean value at the stand level [51]. Changes in growth, mortality or recruitment at the individual level could hardly affect the stable of Dominance. The fact that the mortality or recruitment process impacts the spatial connection between reference trees and adjacent trees at the individual level, but the effect is reduced when computing the mean value of the stand, which may explain why the mean of the Uniform angle index is reasonably constant. When aggregated to the stand average, however, changes in Mingling and Crowding at the individual tree level will not be moderated but will double the effect of promotion, making their stability worse than the Uniform angle index and Dominance. The relationship that the Crowding is less stable than the Mingling at the individual level is naturally extended to the stand level.

*4.3. The Advantages of the New Method and the Limitations in This Study*

Based on the principle of Sankey diagrams, it has strong advantages in displaying the flow among various types or groups in multi-period data [38]. Theoretically, these types or groups can source from any classification variable as long as there are some possible flows between them, like values of SSSPs in this research, trees' group size in a simulated fire study [52], or other attributes in individual level. Therefore, Sankey diagrams can not only show land use change at a large scale, but also have great advantages in displaying stand dynamics at a single tree scale. To further analyze the laws of the stand dynamic changes, each type of flow between two-period was named in this study. The results are greatly helpful to understand the formation mechanism of spatial structure. The analytical method for spatial structure dynamics used in this paper should be popularized in future studies for precise management of the forest structure [53].

We used field data from dynamic monitoring to examine changes in SSSPs at the individual and stand levels, and we attempted to explain the changes using the creative process of each parameter. Although the findings of the study illustrate the dynamic changes laws of the spatial structure of this forest community in this setting, various elements, such as stand type and beginning condition, can impact the changing laws of stand spatial structure [35], the presence of tree pests or diseases can also affect the outcomes of tree mortality if the same species are clustered [50]. The generality of the findings must be verified in other forest communities or simulated by computer programs.

## 5. Conclusions

Multi-period data can evaluate the dynamics of stand spatial structure, whereas previous researches solely focused on comparative assessments of the spatial structure of multi-period live trees, without an in-depth investigation of the change processes. This study attempts to disentangle the dynamics of spatial structure into the flow of live trees, the process of mortality and the process of recruitment at the individual level. The statistics of each "change type" in the three processes were analyzed. Our results show the proportion of changes in the SSSPs of individuals was relatively high, even though the mean values of the stand did not change considerably. The five values (0, 0.25, 0.5, 0.75, 1) of the SSSPs are in mutual flow, and the flows are typically one-step, with three-steps and four-steps changes being uncommon. The spatial structure of fir, maple, ash and spruce in the spruce-fir-broadleaf mixed forest of our study should be improved based on the Crowding, Mingling and Uniform angle index. Our findings also emphasize the impact of the mortality and recruitment processes on spatial structure. The time series can incorporate the spatial structure of a tree as a chain value (change type) rather than limiting it to a single value. These types are highly useful for investigating the changes and stability of SSSPs at the individual, species, and stand level, as well as guiding forest management in some ways. The dynamic change of spatial structure analysis method created in this study can capture more features not discovered in earlier approaches. Understanding the nuances of these changes is an important aspect of maintaining an acceptable spatial structure and biodiversity, and it should be the focus of future study efforts.

**Author Contributions:** Conceptualization, C.Z. and H.Z.; methodology, C.Z. and X.L.; software, C.Z. and L.F.; validation, D.L., K.C. and X.H.; formal analysis, D.L., Y.Z. and C.Z.; investigation, C.Z., K.C. and X.H.; resources, H.Z.; data curation, C.Z. and X.L.; writing—original draft preparation, C.Z. and D.L.; writing—review and editing, K.C., X.H., Y.Z. and H.Z.; visualization, C.Z. and L.F.; supervision, H.Z.; project administration, H.Z.; funding acquisition, H.Z. All authors have read and agreed to the published version of the manuscript.

**Funding:** This research was funded by the Thirteenth Five-year Plan Pioneering project of the High Technology Plan of the National Department of Technology (No.2017YFC0504101).

**Institutional Review Board Statement:** Not applicable.

**Informed Consent Statement:** Not applicable.

**Data Availability Statement:** Data sharing is not applicable.

**Acknowledgments:** We would like to thank all people contributing to sample plot survey for this study. Moreover, we would like to thank the anonymous reviewers and editor for their relevant suggestions on the manuscript.

**Conflicts of Interest:** The authors declare no conflict of interest.

## Appendix A

**Table A1.** The proportion statistics of change types of main tree species in living trees' flow (%).

| Parameters | Types | White Birch | Linden | Ribbed Birch | Korean Pine | Fir | Larch | Maple | Ash | Poplar | Elm | Spruce |
|---|---|---|---|---|---|---|---|---|---|---|---|---|
| Uniform Angle Index | Flow + 4 | 0.0 | 0.0 | 0.0 | 0.0 | 0.0 | 0.0 | 0.0 | 0.0 | 0.0 | 0.0 | 0.0 |
| | Flow + 3 | 0.0 | 0.0 | 0.0 | 0.0 | 0.0 | 0.0 | 0.0 | 0.0 | 0.0 | 0.0 | 0.0 |
| | Flow + 2 | 1.9 | 1.6 | 2.4 | 1.2 | 1.9 | 1.4 | 3.5 | 2.3 | 0.0 | 3.1 | 1.1 |
| | Flow + 1 | 0.0 | 6.0 | 7.7 | 12.0 | 9.4 | 6.2 | 11.2 | 10.5 | 16.0 | 7.7 | 7.7 |
| | Flow 0 | 81.5 | 84.0 | 79.0 | 78.4 | 78.3 | 86.3 | 72.9 | 80.2 | 64.0 | 76.9 | 81.3 |
| | Flow − 1 | 13.0 | 6.3 | 8.6 | 7.2 | 8.2 | 5.3 | 8.8 | 5.8 | 12.0 | 12.3 | 9.9 |
| | Flow − 2 | 3.7 | 2.2 | 2.4 | 1.2 | 2.2 | 0.8 | 3.5 | 1.2 | 8.0 | 0.0 | 0.0 |
| | Flow − 3 | 0.0 | 0.0 | 0.0 | 0.0 | 0.0 | 0.0 | 0.0 | 0.0 | 0.0 | 0.0 | 0.0 |
| | Flow − 4 | 0.0 | 0.0 | 0.0 | 0.0 | 0.0 | 0.0 | 0.0 | 0.0 | 0.0 | 0.0 | 0.0 |
| Dominance | Flow + 4 | 0.0 | 0.0 | 0.0 | 0.0 | 0.0 | 0.0 | 0.0 | 0.0 | 0.0 | 0.0 | 0.0 |
| | Flow + 3 | 0.0 | 0.0 | 0.0 | 0.0 | 0.0 | 0.0 | 0.0 | 0.0 | 0.0 | 0.0 | 0.0 |
| | Flow + 2 | 0.0 | 0.8 | 0.6 | 0.6 | 0.3 | 0.8 | 0.6 | 0.0 | 0.0 | 0.0 | 1.1 |
| | Flow + 1 | 5.6 | 4.9 | 8.0 | 8.4 | 5.7 | 5.9 | 9.4 | 5.8 | 0.0 | 4.6 | 6.6 |
| | Flow 0 | 92.6 | 73.4 | 78.7 | 74.9 | 62.3 | 80.4 | 69.4 | 77.9 | 84.0 | 72.3 | 83.5 |
| | Flow − 1 | 1.9 | 17.7 | 11.5 | 15.0 | 25.8 | 12.0 | 14.7 | 14.0 | 16.0 | 16.9 | 8.8 |
| | Flow − 2 | 0.0 | 3.3 | 1.2 | 0.6 | 5.7 | 0.8 | 5.3 | 2.3 | 0.0 | 6.2 | 0.0 |
| | Flow − 3 | 0.0 | 0.0 | 0.0 | 0.6 | 0.3 | 0.0 | 0.6 | 0.0 | 0.0 | 0.0 | 0.0 |
| | Flow − 4 | 0.0 | 0.0 | 0.0 | 0.0 | 0.0 | 0.0 | 0.0 | 0.0 | 0.0 | 0.0 | 0.0 |
| Mingling | Flow + 4 | 0.0 | 0.0 | 0.0 | 0.0 | 0.0 | 0.0 | 0.0 | 0.0 | 0.0 | 0.0 | 0.0 |
| | Flow + 3 | 0.0 | 0.0 | 0.0 | 0.0 | 0.0 | 0.0 | 0.0 | 0.0 | 0.0 | 0.0 | 0.0 |
| | Flow + 2 | 1.9 | 1.1 | 0.0 | 0.0 | 0.6 | 1.1 | 0.0 | 0.0 | 0.0 | 0.0 | 0.0 |
| | Flow + 1 | 1.9 | 5.2 | 5.9 | 4.2 | 10.7 | 8.7 | 2.9 | 2.3 | 0.0 | 6.2 | 1.1 |
| | Flow 0 | 96.3 | 87.8 | 90.8 | 92.2 | 76.1 | 87.1 | 88.8 | 94.2 | 100.0 | 90.8 | 96.7 |
| | Flow − 1 | 0.0 | 5.4 | 3.3 | 3.6 | 10.1 | 2.8 | 8.2 | 3.5 | 0.0 | 3.1 | 2.2 |
| | Flow − 2 | 0.0 | 0.5 | 0.0 | 0.0 | 2.2 | 0.3 | 0.0 | 0.0 | 0.0 | 0.0 | 0.0 |
| | Flow − 3 | 0.0 | 0.0 | 0.0 | 0.0 | 0.3 | 0.0 | 0.0 | 0.0 | 0.0 | 0.0 | 0.0 |
| | Flow − 4 | 0.0 | 0.0 | 0.0 | 0.0 | 0.0 | 0.0 | 0.0 | 0.0 | 0.0 | 0.0 | 0.0 |
| Crowding | Flow + 4 | 0.0 | 0.0 | 0.0 | 0.6 | 0.0 | 2.5 | 0.0 | 0.0 | 0.0 | 0.0 | 0.0 |
| | Flow + 3 | 0.0 | 0.3 | 0.3 | 1.8 | 0.3 | 7.8 | 0.6 | 1.2 | 0.0 | 1.5 | 1.1 |
| | Flow + 2 | 5.6 | 2.2 | 3.8 | 3.6 | 3.8 | 16.8 | 6.5 | 9.3 | 0.0 | 9.2 | 5.5 |
| | Flow + 1 | 16.7 | 6.5 | 10.9 | 12.0 | 12.6 | 22.4 | 11.2 | 16.3 | 8.0 | 13.8 | 9.9 |
| | Flow 0 | 70.4 | 82.9 | 77.2 | 65.9 | 64.2 | 39.5 | 62.9 | 72.1 | 88.0 | 61.5 | 67.0 |
| | Flow − 1 | 7.4 | 6.0 | 5.9 | 11.4 | 13.8 | 9.2 | 14.1 | 1.2 | 4.0 | 9.2 | 12.1 |
| | Flow − 2 | 0.0 | 1.6 | 1.5 | 4.2 | 4.7 | 1.1 | 2.9 | 0.0 | 0.0 | 4.6 | 4.4 |
| | Flow − 3 | 0.0 | 0.3 | 0.3 | 0.6 | 0.6 | 0.6 | 1.8 | 0.0 | 0.0 | 0.0 | 0.0 |
| | Flow − 4 | 0.0 | 0.3 | 0.0 | 0.0 | 0.0 | 0.0 | 0.0 | 0.0 | 0.0 | 0.0 | 0.0 |
| Number of Trees | | 54 | 368 | 338 | 167 | 318 | 357 | 170 | 86 | 25 | 65 | 91 |

Note: The proportions of each species are calculated by the sum of all three plots. Main species: White birch (*B. platyphylla*); Linden (*T. amurensis*); Ribbed birch (*B. costata*); Korean pine (*P. koraiensis*); Fir (*A. nephrolepis*); Larch (*L. olgensis*); Maple (*A. pictum* subsp. *mono*); Ash (*F. mandschurica*); Poplar (*P. ussuriensis*); Elm (*U. laciniata*); Spruce (*P. jezoensis*). The same is below.

**Table A2.** The proportion statistics of change types of main tree species in mortality process (%).

| Parameters | Types | White Birch | Linden | Ribbed Birch | Korean Pine | Fir | Larch | Maple | Ash | Poplar | Elm | Spruce |
|---|---|---|---|---|---|---|---|---|---|---|---|---|
| Uniform Angle Index | Dead 0 | 25.0 | 0.0↓ | 0.0↓ | 0.0↓ | 0.0↓ | 5.3 | 0.0↓ | 0.0↓ | 0.0 | 0.0 | 0.0 |
| | Dead − 1 | 50.0 | 30.8 | 12.5↓ | 15.4↓ | 15.7↓ | 21.1 | 0.0↓ | 40.0 | 0.0↓ | 25.0 | 33.3 |
| | Dead − 2 | 25.0↓ | 61.5 | 75.0 | 61.5 | 58.8↓ | 36.8↓ | 66.7 | 60.0 | 50.0↓ | 66.7 | 46.7↓ |
| | Dead − 3 | 0.0↓ | 7.7↓ | 6.3↓ | 7.7↓ | 17.6↓ | 26.3 | 0.0↓ | 0.0↓ | 50.0 | 8.3↓ | 13.3↓ |
| | Dead − 4 | 0.0↓ | 0.0↓ | 6.3 | 15.4 | 7.8 | 10.5 | 33.3 | 0.0↓ | 0.0↓ | 0.0↓ | 6.7 |
| Dominance | Dead 0 | 0.0↓ | 0.0↓ | 0.0↓ | 7.7↓ | 21.6 | 21.1↓ | 0.0↓ | 40.0 | 25.0 | 0.0↓ | 60.0 |
| | Dead − 1 | 0.0↓ | 7.7↓ | 18.8↓ | 7.7↓ | 23.5 | 36.8 | 0.0↓ | 0.0↓ | 50.0 | 16.7↓ | 6.7↓ |
| | Dead − 2 | 75.0 | 30.8 | 6.3↓ | 15.4↓ | 23.5 | 15.8↓ | 0.0↓ | 0.0↓ | 0.0↓ | 16.7↓ | 26.7 |
| | Dead − 3 | 0.0↓ | 30.8 | 50.0 | 23.1 | 11.8↓ | 21.1 | 33.3 | 20.0 | 25.0 | 33.3 | 6.7↓ |
| | Dead − 4 | 25.0 | 30.8 | 25.0 | 46.2 | 19.6↓ | 5.3↓ | 66.7 | 40.0 | 0.0↓ | 33.3↓ | 0.0↓ |
| Mingling | Dead 0 | 0.0↓ | 7.7 | 6.3 | 0.0 | 0.0↓ | 10.5 | 0.0↓ | 0.0 | 0.0 | 0.0 | 0.0 |
| | Dead − 1 | 0.0↓ | 23.1 | 0.0↓ | 0.0↓ | 5.9 | 21.1 | 0.0↓ | 0.0↓ | 0.0 | 0.0↓ | 0.0 |
| | Dead − 2 | 25.0 | 23.1 | 25.0 | 15.4 | 33.3 | 21.1 | 0.0↓ | 20.0 | 0.0↓ | 41.7 | 0.0↓ |
| | Dead − 3 | 25.0 | 38.5 | 37.5 | 38.5 | 37.3 | 21.1↓ | 66.7 | 40.0 | 0.0↓ | 16.7↓ | 20.0 |
| | Dead − 4 | 50.0↓ | 7.7↓ | 31.3↓ | 46.2↓ | 23.5↓ | 26.3↓ | 33.3↓ | 40.0↓ | 100.0 | 41.7↓ | 80.0 |

**Table A2.** *Cont.*

| Parameters | Types | White Birch | Linden | Ribbed Birch | Korean Pine | Fir | Larch | Maple | Ash | Poplar | Elm | Spruce |
|---|---|---|---|---|---|---|---|---|---|---|---|---|
| Crowding | Dead 0 | 0.0↓ | 0.0↓ | 6.3 | 0.0↓ | 3.9 | 5.3↓ | 0.0↓ | 0.0↓ | 0.0↓ | 0.0↓ | 0.0↓ |
| | Dead − 1 | 25.0 | 0.0↓ | 12.5 | 7.7 | 2.0↓ | 31.6 | 0.0↓ | 0.0↓ | 0.0↓ | 8.3 | 6.7 |
| | Dead − 2 | 0.0↓ | 7.7 | 31.3 | 7.7↓ | 13.7 | 36.8 | 0.0↓ | 40.0 | 0.0↓ | 8.3↓ | 6.7↓ |
| | Dead − 3 | 25.0 | 7.7↓ | 12.5↓ | 23.1 | 21.6 | 0.0↓ | 33.3 | 20.0 | 50.0 | 16.7↓ | 40.0 |
| | Dead − 4 | 50.0↓ | 84.6 | 37.5↓ | 61.5↓ | 58.8↓ | 26.3↓ | 66.7↓ | 40.0 | 50.0↓ | 66.7 | 46.7↓ |
| Number of Trees | | 4 | 13 | 16 | 13 | 51 | 19 | 3 | 5 | 4 | 12 | 15 |

Note: "↓" indicated that the proportion of this change type is smaller than corresponding values in 2013.

**Table A3.** The proportion statistics of change types of main tree species in recruitment process (%).

| Parameters | Types | White Birch | Linden | Ribbed Birch | Korean Pine | Fir | Larch | Maple | Ash | Poplar | Elm | Spruce |
|---|---|---|---|---|---|---|---|---|---|---|---|---|
| Uniform Angle Index | Reg 0 | 0.0 | 0.0 | 0.0 | 0.0 | 0.0 | 0.0 | 0.0 | 0.0 | 0.0 | 0.0 | 0.0 |
| | Reg + 1 | 0.0 | 18.8↑ | 0.0 | 33.3↑ | 18.8↑ | 100.0↑ | 13.0 | 0.0 | 0.0 | 11.1 | 0.0 |
| | Reg + 2 | 0.0 | 56.3 | 0.0 | 66.7↑ | 50.0 | 0.0 | 60.9↑ | 0.0 | 0.0 | 66.7↑ | 100.0↑ |
| | Reg + 3 | 0.0 | 25.0↑ | 0.0 | 0.0 | 21.4↑ | 0.0 | 17.4 | 100.0↑ | 0.0 | 22.2↑ | 0.0 |
| | Reg + 4 | 0.0 | 0.0 | 0.0 | 0.0 | 9.8↑ | 0.0 | 8.7 | 0.0 | 0.0 | 0.0 | 0.0 |
| Dominance | Reg 0 | 0.0 | 0.0 | 0.0 | 0.0 | 0.0 | 0.0 | 0.0 | 0.0 | 0.0 | 0.0 | 0.0 |
| | Reg + 1 | 0.0 | 0.0 | 0.0 | 0.0 | 0.9 | 0.0 | 0.0 | 0.0 | 0.0 | 0.0 | 0.0 |
| | Reg + 2 | 0.0 | 6.3 | 0.0 | 0.0 | 10.7 | 0.0 | 0.0 | 0.0 | 0.0 | 0.0 | 0.0 |
| | Reg + 3 | 0.0 | 25.0↑ | 0.0 | 11.1 | 24.1↑ | 0.0 | 17.4 | 0.0 | 0.0 | 33.3↑ | 0.0 |
| | Reg + 4 | 0.0 | 68.8↑ | 0.0 | 88.9↑ | 64.3↑ | 100.0↑ | 82.6↑ | 100.0↑ | 0.0 | 66.7↑ | 100.0↑ |
| Mingling | Reg 0 | 0.0 | 0.0 | 0.0 | 0.0 | 3.6↑ | 0.0 | 4.3 | 0.0 | 0.0 | 0.0 | 0.0 |
| | Reg + 1 | 0.0 | 25.0↑ | 0.0 | 0.0 | 9.8↑ | 0.0 | 4.3 | 0.0 | 0.0 | 0.0 | 0.0 |
| | Reg + 2 | 0.0 | 18.8 | 0.0 | 0.0 | 25.0↑ | 100.0↑ | 4.3 | 0.0 | 0.0 | 0.0 | 0.0 |
| | Reg + 3 | 0.0 | 31.3 | 0.0 | 33.3↑ | 31.3 | 0.0 | 30.4↑ | 100.0↑ | 0.0 | 44.4↑ | 0.0 |
| | Reg + 4 | 0.0 | 25.0 | 0.0 | 66.7↑ | 30.4 | 0.0 | 56.5↑ | 0.0 | 0.0 | 55.6 | 100.0↑ |
| Crowding | Reg 0 | 0.0 | 0.0 | 0.0 | 0.0 | 2.7↑ | 0.0 | 0.0 | 0.0 | 0.0 | 0.0 | 0.0 |
| | Reg + 1 | 0.0 | 0.0 | 0.0 | 0.0 | 8.9↑ | 0.0 | 13.0↑ | 0.0 | 0.0 | 0.0 | 33.3↑ |
| | Reg + 2 | 0.0 | 12.5↑ | 0.0 | 0.0 | 19.6↑ | 0.0 | 8.7 | 100.0↑ | 0.0 | 0.0 | 0.0 |
| | Reg + 3 | 0.0 | 12.5↑ | 0.0 | 44.4↑ | 19.6 | 100.0↑ | 30.4↑ | 0.0 | 0.0 | 22.2↑ | 0.0 |
| | Reg + 4 | 0.0 | 75.0 | 0.0 | 55.6 | 49.1 | 0.0 | 47.8 | 0.0 | 0.0 | 77.8↑ | 66.7↑ |
| Number of Trees | | 0 | 16 | 0 | 9 | 112 | 1 | 23 | 1 | 0 | 9 | 3 |

Note: "↑" indicated that the proportion of this change type is larger than corresponding values in 2018.

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
