# Peer review of "Spatial Structure Dynamics and Maintenance of a Natural Mixed Forest"

_forests, doi:10.3390/f13060888_

Round 1

Reviewer 1 Report

The article “Spatial structure dynamics and maintenance of a natural mixed forest” contributed to a new perspective on forest management. Although the approach used may be somewhat restrictive for many investigators, the article is well structured and suitable for publication in the journal Forests.

The Introduction is well developed, presents concrete cases and correctly defines the objectives of the study.

In Methodology, the quality of Figure 1 should be improved. Location names are poorly readable. Can authors create more contrast or make the letters bold?

The selected methods seem to be adequate to the study objectives. However, I miss a better characterization of the studied forest structure. In this topic or in the results I would like to know which species were worked on (since it is a mixed forest) in order to better relate the results obtained.

The authors present the results quite objectively, but I would like to know more about the relationship between the species. It seems to me that the second objective (understand the laws in dynamic changes of spatial structure for spruce-fir-broadleaf mixed forest) is unclear.

The Conclusions are quite general and although they are supported by the results, they do not present surprising results or different from other studies.

Finally, I think that the authors use a lot of Mathematics to give rise to negligible results. What possible implications do these results have for forest management? How could the existence of tree pests or diseases be altering the results of tree mortality?

Reviewer 2 Report

 The manuscript entitled “Spatial structure dynamics and maintenance of a natural mixed forest” is a very interesting example of stand structure analysis using spatial indices based on the nearest neighbourhood of individual trees. The paper proposes a very interesting way of analyzing such data and tests it on real examples of mixed stands. The paper provides a concise introduction to the topic, indicating the potential use of stand spatial structure parameters and their implications for management and biodiversity. The authors clearly explain the proposed methods. It is worth emphasizing that this part of the work is very clearly laid out and allows for easy use of the proposed methodology in other studies. The results and discussion show the changes in the spatial structure of the forest focusing on capturing their causes and highlighting the advantages of the methodology. I believe that the presented article has a great scientific value and deserves to be published in the journal Forests.

Below I present two minor comments, which have the character of discussion with the authors and suggestions of minor changes. They are certainly not important objections.

Line 185-188; 193-195; 201-204. These excerpts deal with a similar aspect of the proposed methodology. It is possible to collect these comments together and present them as one or two sentences at the end of Section 2.3. 

Figure 5: This is a very interesting graph showing changes over time in the spatial structure of the study stands. It seems interesting whether the changes observed over 5 years are statistically significant.  Five years is a very short period in the life of a stand. If the changes are significant it will emphasize the value of the applied methodology. If not it will also provide interesting information and will not undermine the value of the methodology. It seems that a contingency table with the number of trees in each of the five possible values (0, 0.25, 0.5, 0.75 and 1) in 2013 and 2018 and a Chi square test might be a good place to start.  An asterisk denoting significant difference between years for each of the four indicators examined in three stands will enrich this graph.

Round 2

Reviewer 1 Report

The authors read and responded appropriately to the comments made, which resulted in a significant improvement of the manuscript.

The scientific names of the plants must be accompanied by the respective classifier. Classifiers must be introduced at least the first time the botanical name appears in the text.

In the Conclusions the authors must include the main results obtained.

After this small correction, I consider that the article "Spatial structure dynamics and maintenance of a natural mixed forest" can be published in the journal Forests.
